# Magnetodisc modelling in Jupiter's magnetosphere using Juno magnetic field data and the paraboloid magnetic field model

Ivan A. Pensionerov[1], Elena S. Belenkaya[1], Stanley W. H. Cowley[2], Igor I. Alexeev[1], Vladimir V. Kalegaev[1], and David A. Parunakian[1]

[1]Federal State Budget Educational Institution of Higher Education M.V. Lomonosov Moscow State University, Skobeltsyn Institute of Nuclear Physics (SINP MSU), 1(2), Leninskie gory, GSP-1, Moscow 119991, Russian Federation
[2]Department of Physics & Astronomy, University of Leicester, Leicester LE1 7RH, UK

**Correspondence:** I.A. Pensionerov (pensionerov@gmail.com)

**Abstract.** One of the main features of Jupiter's magnetosphere is its equatorial magnetodisc, which significantly increases the field strength and size of the magnetosphere. Analysis of Juno measurements of the magnetic field during the first ten orbits covering the dawn to pre-dawn sector of the magnetosphere ($\sim$3.5–6 hours local time) have allowed us to determine optimal parameters of the magnetodisc using the paraboloid magnetospheric magnetic field model, which employs analytic expressions for the magnetospheric current systems. Specifically within the model we determine the size of the Jovian magnetodisc and the magnetic field strength at its outer edge.

## 1 Introduction

In this paper we consider magnetic field measurements made by the Juno spacecraft in Jupiter's magnetosphere, paying particular attention to the middle magnetosphere measurements where Jupiter's magnetodisc field plays a major role. The structure and properties of the Jovian magnetodisc have been described in many papers starting from the first spacecraft flybys of Jupiter, discussed, e.g., by Barbosa et al. (1979), and references therein. In particular, the empirical magnetodisc model presented by Connerney et al. (1981), derived from Voyager-1 and -2 and Pioneer-10 observations, has been employed as a basis in numerous subsequent studies, including predictions for the Juno mission by Cowley et al. (2008, 2017). Detailed physical models have also been constructed by Caudal (1986), who derived a steady-state MHD magnetodisc model in which both centrifugal and plasma pressure (assumed isotropic) forces were included, and by Nichols (2011) who incorporated a self-consistent plasma angular velocity model. Nichols et al. (2015) have also included the effects of plasma pressure anisotropy, as observed in Voyager and Galileo particle measurements, which redistributes the azimuthal currents in the magnetodisc, changing its thickness.

Here we model the magnetic field observations during Juno's first ten orbits for which both inbound and outbound passes are presently available, corresponding to perijoves (PJs) 0 to 9, using the semi-empirical global paraboloid Jovian magnetospheric

magnetic field model derived by Alexeev and Belenkaya (2005). We focus on the middle magnetosphere, observed on these orbits in the dawn to pre-dawn sector of the magnetosphere (~3.5–6 h local time (LT)), for which the magnetodisc provides the main contribution to the magnetospheric magnetic field. In the model, in which the field contributions are calculated using parameterised analytic equations, the magnetodisc is described by a simple thin plane disc lying in the planetary magnetic equatorial plane. We thus search the paraboloid model magnetodisc input parameters to determine the best fit to the Juno measurements. We note that the magnetodisc may be regarded as the most important source of magnetic field in Jupiter's magnetosphere, with a magnetic moment in the model derived by Alexeev and Belenkaya (2005) using Ulysses inbound data, for example, which is 2.6 times the planetary dipole moment. Consequently, the magnetodisc plays a major role in determining the size of the system in its interaction with the solar wind, and is thus an appropriate focus of study using Juno magnetic field data.

## 2  The Jupiter paraboloid model

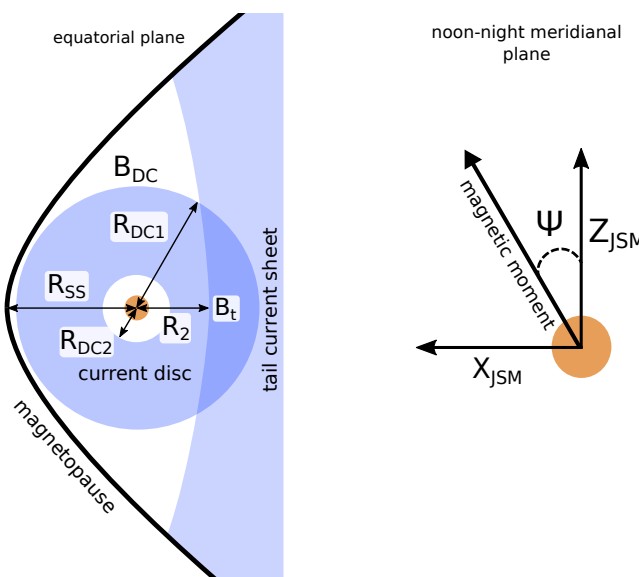

**Figure 1.** On the left we show a schematic of Jupiter's magnetosphere in the magnetic equatorial plane, showing various parameters of the paraboloid model. On the right we show the definition of the planetary magnetic dipole angle $\Psi$ in the JSM system, where $X_{\mathrm{JSM}}$ points towards the Sun and the planetary dipole is contained in the $X_{\mathrm{JSM}}$-$Z_{\mathrm{JSM}}$ plane.

The paraboloid magnetospheric magnetic field model was developed for Jupiter by Alexeev and Belenkaya (2005), based on the terrestrial paraboloid model of Alexeev (1986) and Alexeev et al. (1993). It contains the internal planetary field, $B_{\mathrm{i}}$, calculated from the full order-4 VIP4 model of Connerney et al. (1998), the magnetodisc field, $B_{\mathrm{MD}}$, the field of the magnetopause shielding currents, $B_{\mathrm{si}}$ and $B_{\mathrm{sMD}}$, which screen the planetary and magnetodisc fields, respectively, the field of the magnetotail current system, $B_{\mathrm{TS}}$, and the penetrating part of the interplanetary magnetic field (IMF), $kB_{\mathrm{IMF}}$, where $k$ is

the IMF penetration coefficient. The magnetopause is described by a paraboloid of revolution in Jovian solar magnetospheric (JSM) coordinates with the origin at Jupiter's centre

$$\frac{x}{R_{\mathrm{ss}}} = 1 - \frac{y^2 + z^2}{2R_{\mathrm{ss}}^2} \tag{1}$$

where X is directed towards the Sun, the X-Z plane contains the planet's magnetic moment, and Y completes the right-hand orthogonal set pointing towards dusk. $R_{\mathrm{ss}}$ is the distance to the subsolar magnetopause, where $y = 0$ and $z = 0$. The magnetospheric magnetic field, $\boldsymbol{B}_{\mathrm{m}}$, is then the sum of the fields created by all these current systems

$$\boldsymbol{B}_{\mathrm{m}} = \boldsymbol{B}_{\mathrm{i}}(\Psi) + \boldsymbol{B}_{\mathrm{TS}}(\Psi, R_{\mathrm{ss}}, R_2, B_{\mathrm{t}}) + \boldsymbol{B}_{\mathrm{MD}}(\Psi, B_{\mathrm{DC}}, R_{\mathrm{DC1}}, R_{\mathrm{DC2}}) + \boldsymbol{B}_{\mathrm{si}}(\Psi, R_{\mathrm{ss}}) +$$
$$+ \boldsymbol{B}_{\mathrm{sMD}}(\Psi, R_{\mathrm{ss}}, B_{\mathrm{DC}}, R_{\mathrm{DC1}}, R_{\mathrm{DC2}}) + k\boldsymbol{B}_{\mathrm{IMF}} \tag{2}$$

where $\Psi$ is Jupiter's dipole tilt angle relative to the Z axis. The magnetodisc is approximated as a thin disc with outer and inner radii $R_{\mathrm{DC1}}$ and $R_{\mathrm{DC2}}$, respectively. $B_{\mathrm{DC}}$ is the magnetodisc field at the outer boundary, while the azimuthal currents in the disc are assumed to decrease as $r^{-2}$. $R_2$ is the distance to the inner edge of the tail current sheet, and $B_{\mathrm{t}}$ is the tail current magnetic field there. The magnetospheric current systems are thus described by nine input parameters, determining the physical size of the current systems, and their magnetic field (current) strength ($\Psi$, $R_{\mathrm{ss}}$, $R_2$, $R_{\mathrm{DC1}}$, $R_{\mathrm{DC2}}$, $B_{\mathrm{t}}$, $B_{\mathrm{DC}}$, $k$, $\boldsymbol{B}_{\mathrm{IMF}}$). In Figure 1 we show sketches illustrating the parameters of the model. On the left we show a view in the magnetospheric equatorial plane, where we note that in the physical system, the overlapping model magnetodisc and tail current sheets merge together on the nightside. On the right we show the planetary magnetic dipole axis at angle $\Psi$ in the JSM system. As shown by Alexeev and Belenkaya (2005), the magnetic moment of the model current disc is given by

$$M_{\mathrm{MD}} = \frac{B_{\mathrm{DC}}}{2} R_{\mathrm{DC1}}^3 \left(1 - \frac{R_{\mathrm{DC2}}}{R_{\mathrm{DC1}}}\right). \tag{3}$$

Alexeev and Belenkaya (2005) and Belenkaya (2004) determined model parameters which approximated the magnetic field along the Ulysses inbound trajectory rather well. These parameters are $R_{\mathrm{ss}} = 100\,R_{\mathrm{J}}$, $R_2 = 65\,R_{\mathrm{J}}$, $B_{\mathrm{t}} = -2.5\,\mathrm{nT}$, $R_{\mathrm{DC1}} = 92\,R_{\mathrm{J}}$, $R_{\mathrm{DC2}} = 18.4\,R_{\mathrm{J}}$, and $B_{\mathrm{DC}} = 2.5\,\mathrm{nT}$. This set of parameters is used in the present paper as a starting point for fitting parameters to the Juno data. The dipole tilt angle $\Psi$ changes during the observations and is calculated as a function of time in the paraboloid model.

## 3 Magnetic field calculations for the first ten Juno orbits

As indicated above, field calculations have been made using the paraboloid model for comparison with the data from the first ten Juno orbits for which data are presently available for study. The orbits were closely polar, with large eccentricity, and with apoapsis initially located south of the equator in the dawn magnetosphere (e.g. Connerney et al., 2017).In Figure 2 we show the perijove 1 trajectory versus time (in day of year (DOY) 2016) in JSM Cartesian coordinates, specifically showing the cylindrical and spherical radial distances $\rho_{\mathrm{JSM}} = \sqrt{x^2 + y^2}$ and $r$, $Z_{\mathrm{JSM}}$, and the LT. The vertical dashed line shows the time

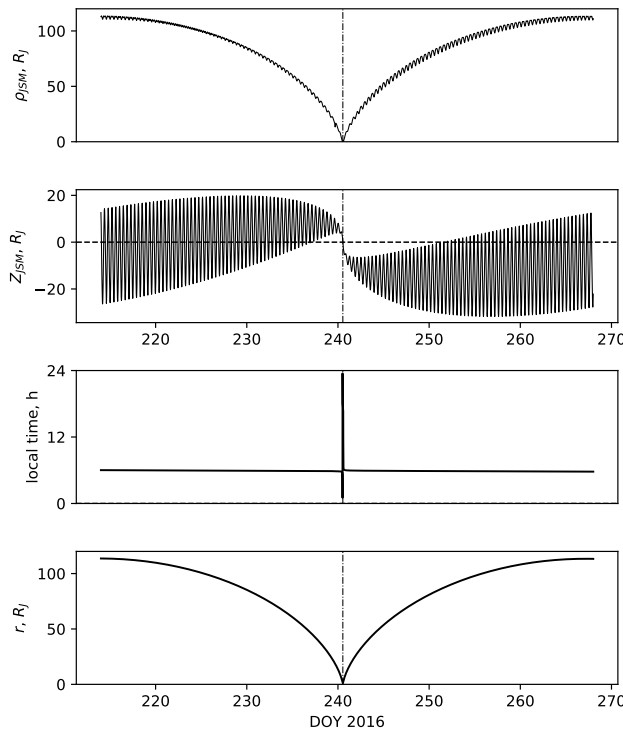

**Figure 2.** Juno perijove 1 trajectory in JSM Cartesian coordinates plotted versus time in DOY 2016, where the vertical dashed line shows the time of periapsis.

of periapsis. On later orbits apoapsis moved towards the nightside reaching $\sim$3.5 h LT by perijove 9, and also rotated further into the southern hemisphere.

In this paper we confine our attention to the middle magnetosphere, where, as we now show, the magnetic field is dominated by the magnetodisc and the planetary field. In the outer magnetosphere the field becomes strongly influenced by external conditions in the solar wind, and although in some circumstances these can be reasonably well predicted by MHD models initialised using data obtained near Earth's orbit (e.g. Tao et al., 2005; Zieger and Hansen, 2008), they will typically vary strongly on the time scale of the Juno orbit (Figure 2), and with them too the outer magnetospheric field. In Figures 3 and 4, for example, we show the magnitudes of the modelled field from different sources along the inbound (left) and outbound (right) passes of perijoves 1 and 9, respectively, plotted versus radial distance. The red lines in these figures show the internal JRM09 ("Juno reference model through perijove 9") planetary field derived by Connerney et al. (2018), which employs the well-determined degree and order 10 coefficients from an overall degree 20 spherical harmonic fit to the data (plus disc model field) from the first nine Juno orbits. The black lines show the field of the various magnetospheric current systems in the paraboloid model as marked, where the model parameters employed are those derived from Ulysses inbound data by Alexeev and Belenkaya (2005), as outlined in Section 2. It can be seen from the figure that for $r < 60\,R_{\mathrm{J}}$ the contributions to the

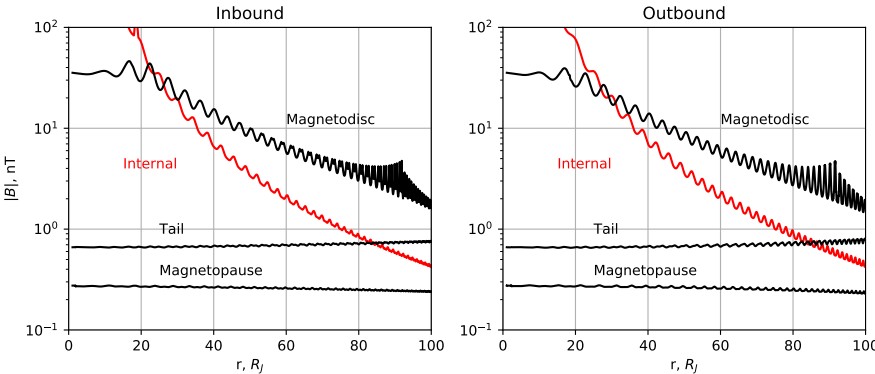

**Figure 3.** Magnitude of the model magnetic fields for the Juno perijove 1 inbound (left) and outbound (right) passes, due to the internal planetary field (JRM09, red), and the various model magnetospheric currents as marked (magnetopause, tail, and magnetodisc, black).

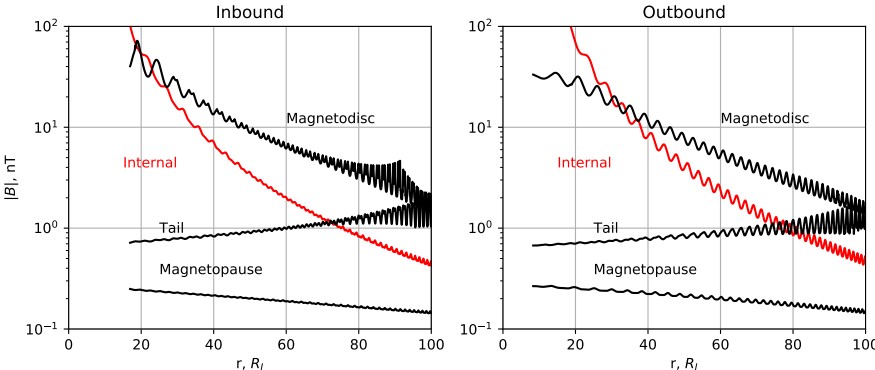

**Figure 4.** As for Figure 3, but for perijove 9.

magnetospheric field from the magnetopause and tail current systems (which are oppositely directed near the dawn-dusk meridian) are negligible compared with the magnetodisc field, being less than 10% for perijove 1 and less than 16% for perijove 9, and may thus be treated approximately inside this distance. For related reasons we also neglect the penetrating IMF term in equation (2), which is unknown when Juno is inside the magnetosphere, highly variable in direction with time, and

5  typically of magnitude ∼0.1-1 nT (Nichols et al., 2006, 2017). This field too, with penetration coefficient $k < 1$, is therefore similarly negligible in the $r < 60\,R_{\mathrm{J}}$ middle magnetosphere studied here.

As a consequence of these considerations, here we employ the JRM09 model of the internal field, and fit only the magnetodisc parameters to the middle magnetosphere data. For the small fields contributed by the magnetopause and tail current systems in this regime, we simply use the Ulysses parameters from Alexeev and Belenkaya (2005) and Belenkaya (2004) as

10  sufficient approximations, i.e., $R_{\mathrm{ss}} = 100\,R_{\mathrm{J}}$, $R_2 = 65\,R_{\mathrm{J}}$, $B_{\mathrm{t}} = -2.5\,\mathrm{nT}$. However, use of the Ulysses magnetodisc parame-

ters is found to lead, for example, to a systematic underestimation of the field along the perijove 1 trajectory, and thus needs to be modified. Thus only three parameters, $R_{\mathrm{DC1}}$, $R_{\mathrm{DC2}}$ and $B_{\mathrm{DC}}$, need to be fitted.

**Table 1.** Magnetodisc parameters derived for the Ulysses inbound pass and the first ten Juno orbits, together with the estimated errors and the minimum inbound and outbound radial distances available in the Juno passes.

| | $B_{\mathrm{DC}}$, nT | $R_{\mathrm{DC2}}$, $R_{\mathrm{J}}$ | $R_{\mathrm{DC1}}$, $R_{\mathrm{J}}$ | $R_{\mathrm{min}}$, $R_{\mathrm{J}}$ inbound | $R_{\mathrm{min}}$, $R_{\mathrm{J}}$ outbound |
|---|---|---|---|---|---|
| Ulysses | 2.50 | 18.4 | 92 | | |
| PJ-00 | $2.58 \pm 0.10$ | $18.7 \pm 2.8$ | 95 | not available | 31.5 |
| PJ-01 | $2.76 \pm 0.12$ | $12.5 \pm 1.8$ | 95 | 5.0 | 5.0 |
| PJ-02 | $2.61 \pm 0.10$ | $13.6 \pm 2.3$ | 95 | 13.3 | not available |
| PJ-03 | $2.79 \pm 0.10$ | $14.5 \pm 1.5$ | 95 | 16.5 | 8.9 |
| PJ-04 | $2.65 \pm 0.07$ | $15.2 \pm 1.2$ | 95 | 13.7 | 12.3 |
| PJ-05 | $2.59 \pm 0.15$ | $14.6 \pm 2.5$ | 95 | 10.6 | 10.5 |
| PJ-06 | $2.70 \pm 0.08$ | $14.3 \pm 1.2$ | 95 | 8.0 | 17.2 |
| PJ-07 | $3.01 \pm 0.09$ | $15.5 \pm 2.0$ | 95 | 21.9 | 19.7 |
| PJ-08 | $3.07 \pm 0.09$ | $15.8 \pm 1.9$ | 95 | 19.5 | 19.5 |
| PJ-09 | $3.06 \pm 0.11$ | $13.6 \pm 1.5$ | 95 | 17.0 | 8.3 |

To optimize the model we choose the approach of minimizing function $S$ given by

$$S(B_{\mathrm{DC}}, R_{\mathrm{DC1}}, R_{\mathrm{DC2}}) = \sqrt{\frac{1}{N} \sum_{n=1}^{N} \frac{\left| \boldsymbol{B}_{\mathrm{mod}}^{(n)} - \boldsymbol{B}_{\mathrm{obs}}^{(n)} \right|^2}{\left| \boldsymbol{B}_{\mathrm{obs}}^{(n)} \right|^2}} \tag{4}$$

5    where $\boldsymbol{B}_{\mathrm{mod}}^{(n)}$ is the modelled field vector due to the current systems, $\boldsymbol{B}_{\mathrm{obs}}^{(n)}$ is the observed residual field following subtraction of the JRM09 internal field model, $n$ is the index number of the data point along the trajectory, and the total number of points is $N$. $S$ represents a root-mean-square relative deviation of the modelled magnetic field from the observed field vectors. We used a relative deviation instead of an absolute value to equalize the influence of all the data points, noting that the magnetic field varies in magnitude significantly along the part of the trajectory examined here (see Figures 3 and 4). Use of the absolute

10    deviation gives good results in the region closer to the planet where the field magnitude is greater, but a poorer fit on other parts of the trajectory.

With regard to the choice of interval employed to minimize $S$, we note that use of data from the innermost region is not optimal. The JRM09 internal planetary field model differs from observations at periapsis ($1.06\,R_{\mathrm{J}}$) by $0.3 \cdot 10^5\,\mathrm{nT}$ (Connerney et al., 2018), which is reasonable accuracy for describing an observed field of magnitude $\sim 8 \cdot 10^5\,\mathrm{nT}$, but does not allow us

15    to distinguish the magnetodisc field of order $100\,\mathrm{nT}$ on this background. We thus restricted the inner border of the interval to

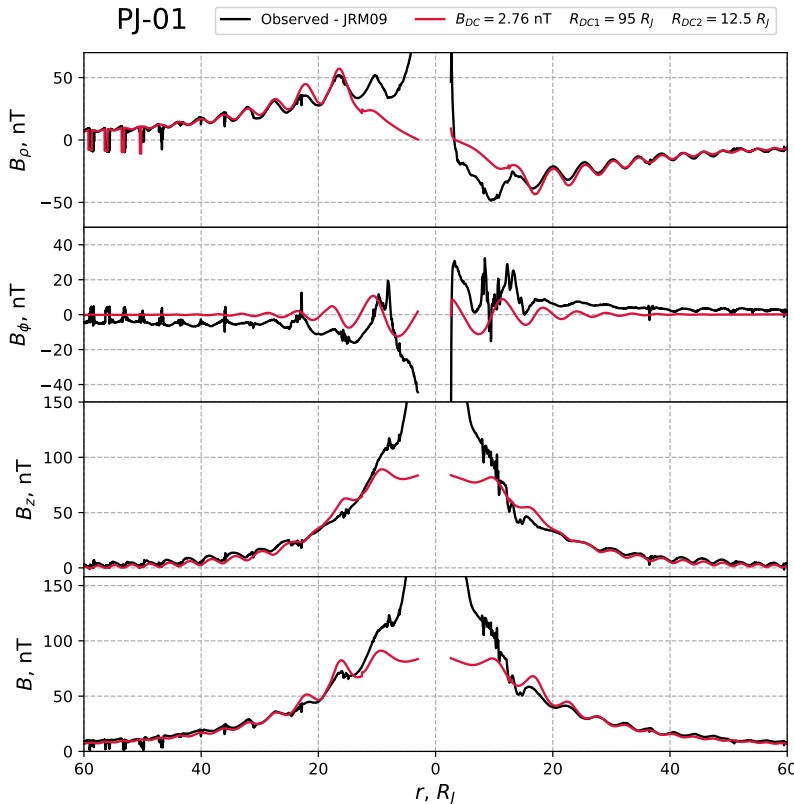

**Figure 5.** Observed (black) and modelled (red) residual fields in JSM cylindrical components, together with the residual field magnitude, for Juno perijove 1. The residual field is the observed field with the JRM09 internal field subtracted. The fields are plotted versus spherical radial distance with inbound data shown on the left and outbound data on the right. The same model field is used for both.

consider $r > 5\,R_{\mathrm{J}}$ only. However, on most passes examined here, the inner radial limit is set instead at somewhat larger radii by the data that are presently available for study. A further limitation on the region of calculation of $S$ in the outer magnetosphere arises from the fact that the paraboloid model does not display regions of low field strength during intersections with the magnetodisc, as is observed in the field at larger distances, due to the use of the infinitely thin disc approximation (see Section 4). It is thus necessary to avoid these regions by excluding parts of the trajectory where the spacecraft is closer than $4\,R_{\mathrm{J}}$ from the magnetic equator.

We thus minimize $S$ in the inbound and outbound radial ranges between $R_{\mathrm{min}}$ and $R_{\mathrm{max}}$ on each pass to determine the best fit magnetodisc parameters. The minimization was undertaken using the Trust Region Reflective procedure (Branch et al., 1999). The best fit values are given, together with the estimated errors values and the radial ranges employed, in Table 1, where we also compare with the values derived by Alexeev and Belenkaya (2005) from Ulysses inbound data. We estimated

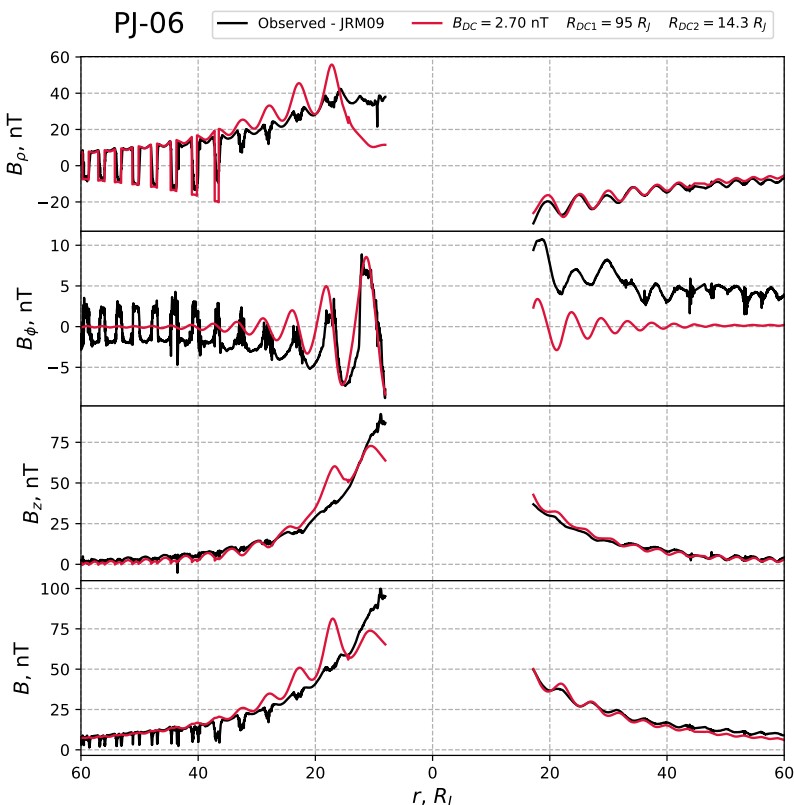

**Figure 6.** As for Figure 5, but for perijove 6.

parameters errors by choosing several different starting points for the algorithm in parameter space and running it with a more generous termination condition in comparison with the normal runs. Specifically, we stopped the calculation when $dS < 0.1S$, where $dS$ is the change of function $S$ in the algorithm step. We then estimated the error as $(P_{\max} - P_{\min})/2$, where $P_{\max}$ and $P_{\min}$ are the maximum and minimum parameter values obtained in these runs. For all the Juno fits we found that the

5  best fit outer disc radius $R_{\mathrm{DC1}}$ was the maximum value of 95 $R_{\mathrm{J}}$ allowed in the fitting process, set by requiring that the disc radius should be less than the subsolar magnetopause radius (100 $R_{\mathrm{J}}$,) by a few $R_{\mathrm{J}}$. This indicates that the current density in the model disc, varying as $r^{-2}$, decreases somewhat too quickly with distance. The values of the inner disc radius $R_{\mathrm{DC2}}$ lie between 12.5 and 18.7 $R_{\mathrm{J}}$, usually smaller than the value of 18.4 $R_{\mathrm{J}}$, derived from the Ulysses data, while the field strength parameter $B_{\mathrm{DC}}$ varies between 2.6 and 3.1 nT, larger than the Ulysses value of 2.5 nT.

10  In Figures 5 and 6 we provide comparisons of the observed (black) and modelled (red) residual fields for Juno perijoves 1 and 6, respectively, from which the JRM09 planetary field has been subtracted. Specifically we show the JSM cylindrical field components together with the residual field magnitude plotted versus radial distance, where the same model applies to both

inbound (left side) and outbound (right side) data. As can be seen, the fitted models are generally in good accordance with the observations for the $B_\rho$ and $B_z$ components, while the $B_\phi$ component is not adequately described, because the model does not include radial currents in the magnetodisc and their closure current via the ionosphere. It is also seen in Figure 5 that the field magnitude is underestimated inside of $\sim$10 $R_\mathrm{J}$, again probably related to the too steep radial dependence of the azimuthal current. As the distance from Jupiter decreases, a sharp increase in the residual field is observed in the inner region to $> 100$ nT, while the model field plateaus at several tens of nT. At the closest distances from the planet the increase is probably due to inaccuracy of the JRM09 model of the internal field, noting that the model represent only the degree and order 10 terms from an overall degree 20 fit (Connerney et al., 2018).

## 4   Approaches for future improvement of the Jupiter paraboloid model

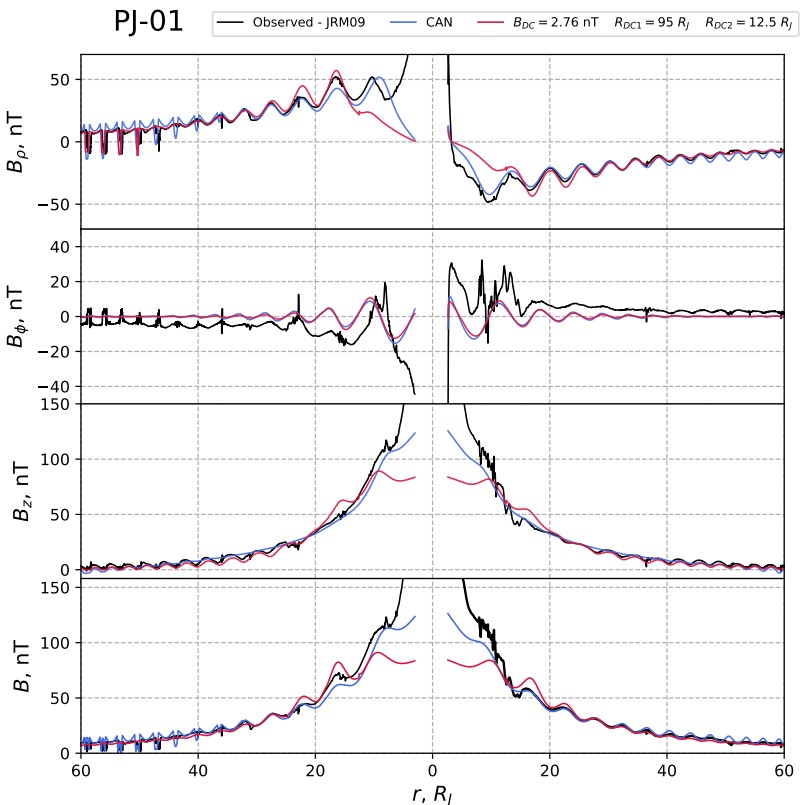

**Figure 7.** Comparison of the observed residual field (black) and best-fit Connerney et al. (1981) magnetodisc model field (blue) in a similar format to Figure 5. We also show the best-fit paraboloid model (red) as in Figure 5.

We first compare the fits derived here with those obtained using the magnetodisc model derived by Connerney et al. (1981) from Voyager-1 and -2 and Pioneer-10 field data, but now fitted to Juno perijove 1 data. In this model the current flows in a planet-centred annular disc of full thickness 5 $R_J$, with inner ($R_0$) and outer ($R_1$) radii at 5 and ∼50 $R_J$, respectively. The azimuthal current in the disc is taken to vary as $I_0/\rho$, where $\rho$ is the perpendicular distance from the planetary dipole magnetic axis. We optimized this model for Juno perijove 1 using the same method as outlined above, to find best-fit parameters $I_0 = 21 \times 10^6 \, A R_J^{-1}$ ($\mu_0 I_0/2 \approx 185$ nT), $R_0 = 6 \, R_J$, and $R_1 = 67 \, R_J$. Figure 7 shows a comparison of the observed residual fields (black) with the best-fit Connerney et al. model (blue) in a similar format to Figures 5 and 6, where we also show the best-fit paraboloid model (red) from Figure 5. One important difference between the model results consists in the fact that the Connerney et al. (1981) model well reflects the observed periodic sharp drops of magnetic field strength during spacecraft intersections with the disc. The magnetodisc radial magnetic field component reverses sign above and below the disc, and at its centre becomes equal to zero. As indicated in Section 3, the paraboloid model having an infinitely thin disc certainly cannot reproduce this feature, and should thus be improved by use of a disc current of finite thickness. The Connerney et al. model demonstrates reasonable coincidence with observations near Jupiter, but at greater distances overestimates the magnetic field strength, which indicates that at these distances the current density variation as $\rho^{-1}$ is too slow.

As indicated above, neither of the magnetodisc models considered here describe the azimuthal field well at medium and large distances, which shows short-term modulations of the field between positive and negative values related to crossings of the current sheet near the planetary rotation period (see, e.g., the inbound data in Figure 6). This points to the well-known existence of radial currents in the magnetodisc associated with sweepback of the field into a "lagging" configuration (e.g. Hill, 1979). Both models considered here, the Connerney et al. (1981) model and the paraboloid model of Alexeev and Belenkaya (2005) do not include these currents, but only the azimuthal current in the magnetodisc. Such radial currents have been included in the models by Khurana (1997) and Cowley et al. (2008, 2017), and could be a useful addition to the paraboloid model, together with their field-aligned and ionospheric closure currents.

## 5 Discussion and Conclusions

As shown in Figures 3 and 4, in the middle part of the Jovian magnetosphere selected for study here, the main contribution to the field due to the magnetospheric current systems is the equatorial magnetodisc. Here we have refined the magnetodisc parameters within the Jovian paraboloid model to best fit the Juno data from the first ten orbits in this region, for which both inbound and outbound data are presently available. Analysis of the field at very close radial distances requires better knowledge of the internal planetary field, while the field at large distances is strongly influenced by the solar wind, whose simultaneous parameters remain unknown and generally varying rapidly with time on the scale of the Juno passes.

As a simplest approximation we took magnetopause and tail current parameters derived using the Ulysses mission data (Alexeev and Belenkaya, 2005; Belenkaya, 2004), and changed only the radial and field strength parameters of the magnetodisc. We found that the best fit model consistently had a large outer radius comparable with the subsolar magnetopause distance

(taken to be 100 $R_J$ from the Ulysses model), an inner radius usually between ∼12 and 14 $R_J$ smaller than the Ulysses model (∼18 $R_J$), and a comparable field strength parameter (at the outer edge of the disc) of ∼2.5 nT.

To further refine the Jovian paraboloid magnetospheric model, it will be necessary to take into account the finite thickness of the magnetodisc current, and also to accurately determine its dependence on the radial distance from the planet. The existence of radial currents in the disc, as well as their closure via field-aligned currents in the planetary ionosphere, should also be incorporated.

*Code availability.* Those who would like to work with the paraboloid model may contact Igor I. Alexeev at alexeev@dec1.sinp.msu.ru.

*Competing interests.* The authors declare that they have no conflict of interest.

*Acknowledgements.* Work at the Federal State Budget Educational Institution of Higher Education M.V. Lomonosov Moscow State University, Skobeltsyn Institute of Nuclear Physics (SINP MSU) was partially supported by the Ministry of Education and Science of the Russian Federation (grant RFMEFI61617X0084). Work at the University of Leicester was supported by STFC grant ST/N000749/1. The Juno magnetometer data were obtained from the Planetary Data System (PDS). We are grateful to the Juno team for making the magnetic field data available (FGM instrument scientist J. E. P. Connerney; principal investigator of Juno mission Scott J. Bolton).

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
