# Peer review of "Magnetodisc modelling in Jupiter's magnetosphere using Juno magnetic field data and the paraboloid magnetic field model"

_Annales Geophysicae, 2018_

## Short Comment (SC1) · 27 Aug 2018

Manuscript is dedicated to the improvement of a "paraboloid model" of the low latitudes middle Jupiter's magnetosphere ($20R_J < r < 60R_J$). Many aspects of this model were examined in the foregoing articles of the authors. An important advantage of the model under consideration is the use in calculations close to real current systems and IMF (in this case IMF is not known, but does not play a significant role). New results based on modern experimental data, obtained within the framework of the Juno magnetic field measurements. The authors is carried out detail comparison of the new results with the earlier studies.

[Figure]

The results obtained by the authors are important for understanding the conditions of plasma convection, details of the equatorial plasma shields formation, radial diffusion in the radiation belts, etc. It is useful for the members of MOP community to be able to use this manuscript as a regular article.

---

## Referee Comment (RC1) · Anonymous Referee #1 · 14 Sep 2018

General comments

The paper adjusts the paraboloid Jovian magnetospheric magnetic field model from Alexeev & Belenkaya 2005 to magnetic field data recorded by Juno in the middle magnetosphere during its first perijove of august 2016. Two of the nine model parameters are constrained by the selected measurements (the magnetodisc inner radius R_DC2, and the magnetodisc field at its outer boundary B_DC), the other seven being fixed at their value deduced from the Ulysses flyby. The new values differ by resp. 14% and 26% from their Ulysses values, the error bars making the new R_DC2 value marginally consistent with the Ulysses one. The authors carefully discuss the selection of the 2

parameters to fit (while retaining the others at their Ulysses values) and the possible future improvements of the paraboloid model.

While the new values of B_DC and R_DC2 may be useful to colleagues working on the magnetosphere of Jupiter, I consider that a fit of 2 parameters from a single Juno perijove (out of 14 up to now) does not justify the publication of a regular article. With further work, there seems to be matter for a good regular article along two possible lines (not mutually exclusive): (1) analyzing many more Juno perijoves and studying the variability of the adjusted parameters, the fit quality, the possibility to constrain more parameters, to perform a global multi-perijove fit, etc. and/or (2) proceeding to some improvements of the paraboloid model (the most obvious one being to replace the infinitely thin disc by one of finite thickness) before applying it to Juno data.

Accordingly, I request a major revision of the present manuscript.

Specific comments

The scientific interest for determining a new fit of some parameters of the paraboloid model is not discussed.

It is not clear if inbound and outbound passes are considered separately in the plots only (e.g. Figs 2, 4, 5), or also for the adjustment. In the latter case, it should be justified and the values found for the 2 legs compared.

The covariance of B_DC and R_DC2 with the other 7 parameters could be better discussed. How are uncertainties likely to be affected ? Would this not imply that the present determinations of B_DC and R_DC2 are actually compatible with Ulysses data ? For example, you state that "deep and sharp field decreases due to the equatorial current sheet encounters continue to be observed on the Juno trajectory even at large radial distances r > 90RJ". May this imply that the Ulysses value of the outer radius of the magnetodisk RDC1 = 92RJ is actually underestimated ?

On p.8, you mention about the upstream solar wind "the limited information obtained

by computer modelling using data from near Earth orbit as input". But there are today very good models of solar wind propagation to Jupiter and beyond (mSWiM model of Zieger & Hansen 2008, or the model from Tao et al. 2005).

Technical corrections

I may be worth saying in the title which part of the magnetosphere is studied (e.g. the magnetodisc) rather than mentioning only the data and the model.

p.1 l.11: flybys OF Jupiter ? (NB: this is only a suggestion, the native english-speaking co-author is certainly more knowledgeable than me about the style)

p.1 l.16: what do you mean by "angular model".

p.2 l.23: a sketch illustrating the 9 parameters would be useful.

p.3 l.11: maybe precise that "negligible" means here "<10% of".

p.3, l.15: explain why "the use of averaged parameters is not adequate in this region", i.e. address the solar wind driven variability.

p.4, l.19: rather than discarding the use of the root-mean-square absolute deviation because it depend strongly on the position of the inner fitting interval boundary, could another option be to use both it (to perhaps better constrain R_DC2) and the relative deviation (for B_DC and R_DC2) ?

Caption of Fig. 4: the JRM09 model has not been subtracted from the residual magnetic field but from the observations.

---

## Referee Comment (RC2) · Anonymous Referee #2 · 31 Oct 2018

**Comments on paper angeo-2018-82**
**"Analysis of Juno perijove 1 magnetic field data using the Jovian paraboloid magnetospheric model" by Pensionerov *et al.**

**1   General comments**

In this paper the authors present Jovian magnetic field measurements from the middle magnetosphere collected during Juno perijove 1 pass. The data are analysed in order to determine optimal parameters for the magnetodisc described by the semi empirical global paraboloid Jovian magnetic field model by Alexeev and Belenkaya (2005). This model consists of six components contributing to the total magnetospheric magnetic field (internal field, IMF and different current systems contributions).

In their analysis, the magnetic field data are kept untouched, and the principal contributions to the magnetic field in the observed region (middle magnetosphere) are assumed to be the internal field and the magnetodisc. Only two parameters of the four parameters to describe the magnetodisc are 'fitted' (while there are a total of nine parameters for the global magnetic field). These parameters are the radius of the inner edge of the disc $R_{DC2}$ and the magnetic field at the outer edge of the magnetodisc $B_{DC}$, the other two parameters consist of Jupiter's dipole $\psi$ (and is calculated as function of time), and the radius of the outer edge of the disc $R_{DC1}$ (fixed to the value given by Alexeev and Belenkaya (2005) with data from the inbound trajectory of Ulysses).

Similar studies to estimate the magnetodisc's parameters according to a model have been carried for Jupiter (as well as Saturn) with empirical models such as the CAN disc (Connerney, Acuna and Ness, 1983) using magnetic data from various missions (Voyager, Pioneer, Galileo, Ulysses, Cassini). There are also detailed physical models such as Caudal (1986), and Achilleos, Guio and Arridge (2010) for Saturn to which magnetic data have been compared. This study is carried using magnetic data collected from the on-going mission to Jupiter, Juno. This could potentially contribute and add to the existing knowledge from previous work but I believe that the article in its present form is not acceptable for publication in Annales Geophysicae. But I would encourage the authors to resubmit their paper after implementing the revisions as proposed hereafter.

**2 Specific comments**

In an age where advanced nonlinear fitting programs and methods have never been so easy to access, I find it somehow not acceptable to 'characterise' the best fit of a multi-parameter fit model with a contour plot of the residuals for the two parameters $B_{DC}$ and $R_{DC2}$ (Fig. 3). I would recommend to try and use a standard nonlinear fitting program implementing a Levenberg Marquardt method or similar, that provides as well meaningful statistics like error estimates for the parameters. You might be want as well to try and fit $R_{DC1}$ with such method.

Eq. 3 does not make sense in its present form. The numerator under the summation over measurement points is homogeneous to the square of a vector while a scalar is meant: the Euclidean vector norm. It is not clear what is actually fitted, the components of the vector (in what coordinate system?). Figures 4, 5 and 6 all show the amplitude of the magnetic field. It would be more meaningful to present the radial, meridional, azimuthal components and the amplitude of the residual magnetic field in order to identify the component that 'best' fit (the radial component?) and 'worst' fit (the azimuthal component due to the poloidal nature of the field/lack of bend-back model?).

In Figures 4 and 5 the observations at large radial distance exhibit large fluctuations. Do you have any explanation?. Does it make sense at all to include these data in the fit? Wouldn't it be better to smooth the data first? Also does Figure 4 (inbound) not suggest that $R_{DC2}$ could be larger than $92R_J$ while in Figure 5 (outbound) $R_{DC2}$ could be smaller than $92R_J$?

In Section 4, I would recommend to carry out a valid and fair comparison with the CAN disc by actually fitting the four parameters of the CAN disc, otherwise the comparison seems arbitrary.

Figures could be made slightly bigger in general. In Figure 1 it might be useful to add panels for cylindrical and spherical distance as well as local time.

**3 Technical corrections**

- l. 27, p. 2: it would be nice to elaborate on why the IMF better off neglected rather than considering a typical value.

- paragraph starting l. 12, p. 4: the discussion on the sensibility of $S$ to the range of measurements considered for the fit is of prime importance. It needs some clarification and also expanded to justify the choice of range.

- l. 7, p. 5: what does the value $S = 0.2$ correspond to concretely in terms of statistics? As it stands it seems to be an arbitrary choice.

- l. 14, p. 5: I am not sure to follow the argument. What is it meant by 'acceptable pairs of parameters are aligned with the line to some extent'? Some clarification needed.

- l. 1-5, p. 6: the discussion about the discrepancies observed in the internal field is too vague and lack content.

- paragraph starting l. 2, p. 7: as mentioned before does it really make sense to take arbitrary values for the CAN disc parameters. Wouldn't it make more sense to carry a proper fit?

- l. 7, p. 8: what do you mean by 'magnetodisc models with azimuthal current dependencies different from $r^{-2}$ should also be investigated'? The CAN disc model just used in that Section varies as $r^{-1}$. Do you have any suggestion? In Achilleos, Guio and Arridge (2010), it is suggested that the dependency is steeper than $r^{-1}$.

---

## Author Comment (AC1) · 28 Nov 2018

The responses to Anonymous Reviewer 1 with revised version of the article, and version with changes marked were packed in .zip archive and uploaded in the form of a supplement. The archive also contains answers to the Anonymous Reviewer 2, in case Anonymous Reviewer 1 wants to read them.

Please also note the supplement to this comment:
https://www.ann-geophys-discuss.net/angeo-2018-82/angeo-2018-82-AC1-supplement.zip

---

## Author Response (AR1)

**Responses to comments on "Analysis of Juno perijove 1 magnetic field data using the Jovian paraboloid magnetospheric model" by Ivan A Pensionerov et al. (Manuscript number angeo-2018-82)**

**Anonymous Referee #1**

We are grateful to the Referee for their comments and attention to our work. The comments are reproduced verbatim in italics, and our replies given step-by-step beneath. The page and line numbers are given for the revised manuscript.

*General comments*

*The paper adjusts the paraboloid Jovian magnetospheric magnetic field model from Alexeev & Belenkaya 2005 to magnetic field data recorded by Juno in the middle magnetosphere during its first perijove of august 2016. Two of the nine model parameters are constrained by the selected measurements (the magnetodisc inner radius R_DC2, and the magnetodisc field at its outer boundary B_DC), the other seven being fixed at their value deduced from the Ulysses flyby. The new values differ by resp. 14% and 26% from their Ulysses values, the error bars making the new R_DC2 value marginally consistent with the Ulysses one. The authors carefully discuss the selection of the 2 parameters to fit (while retaining the others at their Ulysses values) and the possible future improvements of the paraboloid model.*
* * *
The referee correctly describes the content of the present work. However, we emphasize that the reasons for focusing on the selected model parameters while retaining others at the Ulysses values was carefully discussed and justified in the paper, as it is in the revised version. That is to say, in the dawn sector of the middle magnetosphere the role of the magnetodisc is predominant.
* * *
*While the new values of B_DC and R_DC2 may be useful to colleagues working on the magnetosphere of Jupiter, I consider that a fit of 2 parameters from a single Juno perijove (out of 14 up to now) does not justify the publication of a regular article. With further work, there seems to be matter for a good regular article along two possible lines (not mutually exclusive): (1) analyzing many more Juno perijoves and studying the variability of the adjusted parameters, the fit quality, the possibility to constrain more parameters, to perform a global multi-perijove fit, etc. and/or (2) proceeding to some improvements of the paraboloid model (the most obvious one being to replace the infinitely thin disc by one of finite thickness) before applying it to Juno data. Accordingly, I request a major revision of the present manuscript.*
* * *
In response to the referee's comment we have now enhanced the article by analysing data from the first ten Juno perijoves. All of them except PJ-01 lack the near-perijove data in a variable manner, which was the reason to choose to examine PJ-01 specifically in the original article. In addition, we also included all three magnetodisc parameters into the fit (inner and outer radius and field strength parameter), and, in response to comments from Referee 2, also improved the method of model parameter optimization to an automated non-linear optimization procedure (see responses to Referee 2). However, the results show that the best-fit model always has an outer radius at the maximum value set (95 $R_J$) by the model distance of the subsolar magnetopause. As indicated above, the reasons for employing the Ulysses values of the minor field contributions is fully discussed and justified. Overall, however, the article has been significantly revised, with Section 3 undergoing the most important changes. Concerning the comments on improving the paraboloid model, this is clearly outside the scope of the present paper, and is the subject of ongoing and future work. However, the present paper allows us to reveal the points which need improvement, specifically the thickness of the disc and variable dependence of the current density with radial distance.
* * *
*Specific comments*

*The scientific interest for determining a new fit of some parameters of the paraboloid model is not discussed.*
* * *
In response to this comment we have now inserted the following in the Introduction at page 2 lines 6-10, which we feel explains the significance of the study. "We note that the magnetodisc may be regarded as the most important source of magnetic field in Jupiter's magnetosphere, with a magnetic moment in the model derived by Alexeev and Belenkaya (2005) using Ulysses inbound data, for example, which is 2.6 times the planetary dipole moment. Consequently, the magnetodisc plays a major role in determining the size of the system in its interaction with the solar wind, and is thus an appropriate focus of study using Juno magnetic field data."
* * *
*It is not clear if inbound and outbound passes are considered separately in the plots only (e.g. Figs 2, 4, 5), or also for the adjustment. In the latter case, it should be justified and the values found for the 2 legs compared.*
* * *
The model parameters are the same for both legs of each orbit. In the revised paper the inbound and outbound passes are shown in the same figure to make this clear, and it is stated explicitly in the caption of Figure 5.
* * *
*The covariance of B_DC and R_DC2 with the other 7 parameters could be better discussed. How are uncertainties likely to be affected ? Would this not imply that the present determinations of B_DC and R_DC2 are actually compatible with Ulysses data?*
* * *
Evidently the fit results for the magnetodisc parameters could be significantly altered from those given in the paper if, e.g., the tail and magnetopause current parameters were varied through arbitrary ranges. However, as shown in Figures 3 and 4 the fields due to the tail and magnetopause currents in the Ulysses model are at least an order of magnitude less than the field due to the magnetodisc in the region inside 60 $R_J$ considered in the paper, such that they will remain small in any plausibly modified model. This conclusion is reinforced by the fact, now noted in the related text, that the tail and magnetopause fields have opposite senses, and hence partly cancel. Brief examination then indicates that if these parameters are varied within plausible ranges, the disc parameters are altered by ~10%. For the purposes of the present paper we therefore believe it to be most satisfactory to compare disc parameters between Juno orbits while holding the minor contributing fields at constant and reasonable values.
* * *
*For example, you state that "deep and sharp field decreases due to the equatorial current sheet encounters continue to be observed on the Juno trajectory even at large radial distances r > 90RJ". May this imply that the Ulysses value of the outer radius of the magnetodisk RDC1 = 92RJ is actually underestimated?*
* * *
We believe the outer radius of the disk is not underestimated in the submitted or revised papers, for the following reason. We have to recognise that in the physical system near the dawn-dusk meridian and on

the nightside the magnetodisc current sheet merges directly into the tail current sheet, so that at large distances it is the tail current sheet that is being observed. In the model, however, the magnetodisc is treated as axisymmetric, with a radius that for physical consistency must be limited to lie a least a little inside the subsolar magnetopause. The continuing current sheet on the nightside is then treated in the model as a separate current system as fully discussed in section 2, and now illustrated in Figure 1 in the revised article. We note that in the revised paper we also treated the outer magnetodisc radius as an adjustable parameter determined from a fit to the data as indicated above, but found that the best-fit value was always the largest value allowed by the above physical restriction, i.e., an outer radius of 95 $R_J$ compared with a subsolar magnetopause radius of 100 $R_J$.
* * *
*On p.8, you mention about the upstream solar wind "the limited information obtained by computer modelling using data from near Earth orbit as input". But there are today very good models of solar wind propagation to Jupiter and beyond (mSWiM model of Zieger & Hansen 2008, or the model from Tao et al. 2005).*
* * *
Despite the acknowledged limitations of solar wind MHD modelling from Earth's orbit into the outer solar system (e.g., requirement for reasonable Earth-planet alignment, uncertainties in arrival time of a day or so, and inability to predict the north-south IMF component), the remarks in the submitted paper on this point were perhaps a little too negative. However, this discussion misses the main point about variability, since the solar wind will typically vary strongly on the time scale of the Juno passes, the overall orbit period being approximately two solar rotations. Such variability makes the task of modelling the field conditions in the outer magnetosphere very challenging, even if one has reasonable knowledge of the input conditions from MHD models. It is for this reason that we focus here on the dawn sector middle magnetosphere inside of ~60 $R_J$ where, as we have indicated, conditions are not strongly influenced by the solar wind-related fields, but are instead dominated by the field of the magnetodisc (plus the planetary field). On page 4 lines 3–7 we have replaced the above comments by the following text, which we believe takes care of the referee's comments.

"In this paper we confine our attention to the middle magnetosphere, where, as we now show, the magnetic field is dominated by the magnetodisc and the planetary field. In the outer magnetosphere the field becomes strongly influenced by external conditions in the solar wind, and although in some circumstances these can be reasonably well predicted by MHD models initialised using data obtained near Earth's orbit (e.g. Tao et al., 2005; Zieger and Hansen, 2008), they will typically vary strongly on the time scale of the Juno orbit (Figure 2), and with them too the outer magnetospheric field."
* * *
*Technical corrections*

*It may be worth saying in the title which part of the magnetosphere is studied (e.g. the magnetodisc) rather than mentioning only the data and the model.*
* * *
In response to this comment we have now changed the title to "Magnetodisc modelling in Jupiter's magnetosphere using Juno magnetic field data and the paraboloid magnetic field model".
* * *
*p.1 l.11: flybys OF Jupiter ? (NB: this is only a suggestion, the native english-speaking co-author is certainly more knowledgeable than me about the style)*
* * *
Changed as suggested (page 1 line 11).
* * *
*p.1 l.16: what do you mean by "angular model".*
* * *
Now corrected to "plasma angular velocity model" (page 1 line 17).
* * *
*p.2 l.23: a sketch illustrating the 9 parameters would be useful.*
* * *
In response to this comment we have added new Figure 1 in the revised version illustrating seven of the model parameters employed. Parameters $k$ and $B_{IMF}$ are not included in the analysis for reasons fully discussed in the paper, and are consequently not shown in the figure.
* * *
*p.3 l.11: maybe precise that "negligible" means here "<10% of».*
* * *
In accordance with the referee's comment the text now specifies "less than 10%" (page 5 lines 2–3).
* * *
*p.3, l.15: explain why "the use of averaged parameters is not adequate in this region», i.e. address the solar wind driven variability.*
* * *
This issue about solar wind and outer magnetosphere variability is fully dealt with above. This specific text is now omitted in the revised paper.
* * *
*p.4, l.19: rather than discarding the use of the root-mean-square absolute deviation because it depend strongly on the position of the inner fitting interval boundary, could another option be to use both it (to perhaps better constrain R_DC2) and the relative deviation (for B_DC and R_DC2) ?*
* * *
This point is now discussed more fully in the revised version on page 6 lines 7-11. Use of the absolute deviation strongly emphasises the fit in the inner region where the residual fields are the largest. We regard the relative deviation as preferable since it equalizes the influence of the data from the whole interval employed, and gives a better fit to the data overall. A comparison of the fits for PJ-01 is shown in the figure attached below.
* * *
*Caption of Fig. 4: the JRM09 model has not been subtracted from the residual magnetic*

*field but from the observations.*
* * *
The caption (now Figure 5) has been revised as follows.

"Observed (black) and modelled (red) residual fields in JSM cylindrical components, together with the residual field magnitude, for Juno perijove 1. The residual field is the observed field with the JRM09

internal field subtracted. The fields are plotted versus spherical radial distance with inbound data shown on the left and outbound data on the right. The same model field is used for both."
* * *
**Ivan A Pensionerov on behalf of the co-authors**

**27 November 2018**

[Figure]

**Responses to comments on "Analysis of Juno perijove 1 magnetic field data using the Jovian paraboloid magnetospheric model" by Ivan A Pensionerov et al. (Manuscript number angeo-2018-82)**

**Anonymous Referee #2**

We are grateful to the Referee for their comments, which have resulted in a number of significant changes in the revised version. The comments are reproduced verbatim in italics, and our replies given step-by-step beneath. The page and line numbers are given for the revised manuscript.

*General comments*

*In this paper the authors present Jovian magnetic field measurements from the middle magneto- sphere collected during Juno perijove 1 pass. The data are analysed in order to determine optimal parameters for the magnetodisc described by the semi empirical global paraboloid Jovian magnetic field model by Alexeev and Belenkaya (2005). This model consists of six components contributing to the total magnetospheric magnetic field (internal field, IMF and different current systems contributions).*

*In their analysis, the magnetic field data are kept untouched, and the principal contributions to the magnetic field in the observed region (middle magnetosphere) are assumed to be the internal field and the magnetodisc. Only two parameters of the four parameters to describe the magnetodisc are 'fitted' (while there are a total of nine parameters for the global magnetic field). These parameters are the radius of the inner edge of the disc RDC2 and the magnetic field at the outer edge of the magnetodisc BDC, the other two parameters consist of Jupiter's dipole ψ (and is calculated as function of time), and the radius of the outer edge of the disc RDC1 (fixed to the value given by Alexeev and Belenkaya (2005) with data from the inbound trajectory of Ulysses).*

*Similar studies to estimate the magnetodisc's parameters according to a model have been carried for Jupiter (as well as Saturn) with empirical models such as the CAN disc (Connerney, Acuna and Ness, 1983) using magnetic data from various missions (Voyager, Pioneer, Galileo, Ulysses, Cassini). There are also detailed physical models such as Caudal (1986), and Achilleos, Guio and Arridge (2010) for Saturn to which magnetic data have been compared. This study is carried using magnetic data collected from the on-going mission to Jupiter, Juno. This could potentially contribute and add to the existing knowledge from previous work but I believe that the article in its present form is not acceptable for publication in Annales Geophysicae. But I would encourage the authors to resubmit their paper after implementing the revisions as proposed hereafter.*
* * *
The referee's description of our paper is mainly correct. We point out, however, that it is shown directly in the paper (Figures 3 and 4 in the revised version) that fields in the regime considered, inside 60 $R_J$, are indeed dominated by the magnetodisc and planetary fields, such that this is not an assumption as stated above. This finding then makes it reasonable to treat the minor field contributions from the tail and magnetopause currents in an approximate way, by using fixed parameter values set at those determined from the Ulysses inbound pass. These fields are typically at least an order of magnitude less than the magnetodisc field in the middle magnetosphere regime investigated, such that plausible modifications will not change the fit to the magnetodisk field significantly.
* * *
*Specific comments*

*In an age where advanced nonlinear fitting programs and methods have never been so easy to access, I find it somehow not acceptable to 'characterise' the best fit of a multi-parameter fit model with a contour plot of the residuals for the two parameters BDC and RDC2 (Fig. 3). I would recommend to try and use a standard nonlinear fitting program implementing a Levenberg Marquardt method or similar, that provides*

*as well meaningful statistics like error estimates for the parameters. You might be want as well to try and fit RDC1 with such method.*
* * *
In response to this comment we have changed the method of parameter optimization to the "Trust Region Reflective" procedure (Branch et al., 1999), as indicated on page 7 lines 9-10. We also newly included $R_{DC1}$ (outer disc radius) into the fit. However, the best $R_{DC1}$ value for all 10 orbits employed in the study was found to be the maximum value set in relation to the size of the model subsolar magnetopause, namely 95 $R_J$ (Table 1 and page 8 lines 1–3).

Branch, M. A., Coleman, T. F., and Li, Y.: A Subspace, Interior, and Conjugate Gradient Method for Large-Scale Bound-Constrained Minimization Problems, SIAM Journal on Scientific Computing, 21, 1–23, https://doi.org/10.1137/s1064827595289108, 1999
* * *
*Eq. 3 does not make sense in its present form. The numerator under the summation over measurement points is homogeneous to the square of a vector while a scalar is meant: the Euclidean vector norm. It is not clear what is actually fitted, the components of the vector (in what coordinate system?).*
* * *
In the revised paper the form of the equation, now Eq (4), has been clarified, and its denominator changed from the magnitude of the modelled field to the magnitude of the observed residual field. The calculation was carried through using Cartesian components in the JSM system, but this is actually immaterial since the vector magnitudes employed are entirely independent of the chosen coordinate system.
* * *
*Figures 4, 5 and 6 all show the amplitude of the magnetic field. It would be more meaningful to present the radial, meridional, azimuthal components and the amplitude of the residual magnetic field in order to identify the component that 'best' fit (the radial component?) and 'worst' fit (the azimuthal component due to the poloidal nature of the field/lack of bend-back model?).*
* * *
In conformity with the referee's comments, in the revised version Figures 5-7 showing the modelling results for the residual field now display cylindrical components in the JSM system together with the field magnitude. All of these figures have been significantly changed during revision.
* * *
*In Figures 4 and 5 the observations at large radial distance exhibit large fluctuations. Do you have any explanation?. Does it make sense at all to include these data in the fit? Wouldn't it be better to smooth the data first? Also does Figure 4 (inbound) not suggest that RDC2 could be larger than 92RJ while in Figure 5 (outbound) RDC2 could be smaller than 92RJ ?*
* * *
We believe the referee is referring to the radius of the outer boundary of the disc, $R_{DC1}$, in the above comments. As discussed in the paper, the field in the outer magnetosphere is strongly influenced by the variable and not well known conditions in the solar wind/IMF. For this reason we restricted our analysis of the dawn sector Juno data to the radial range less than 60 $R_J$ where the relative influence of the solar wind is far less, and the field variations rather smooth. We do not include the fluctuating data at large radial distances into our fit.
* * *
*In Section 4, I would recommend to carry out a valid and fair comparison with the CAN disc by actually fitting the four parameters of the CAN disc, otherwise the comparison seems arbitrary.*
* * *
In response to this comment, we have now carried out a fair comparison of results using the CAN model to fit to the residual data, with results described in Sect 4 and Figure 7.
* * *
*Figures could be made slightly bigger in general. In Figure 1 it might be useful to add panels for cylindrical and spherical distance as well as local time.*
* * *
The figures have been made larger, and panels as suggested have been added to Figure 2 (was Figure 1).
* * *
*Technical corrections*

*l. 27, p. 2: it would be nice to elaborate on why the IMF better off neglected rather than considering a typical value.*
* * *
This issue is now discussed in more detail on page 5 lines 3–6. Basically, the added field would be small, of order the tail and magnetopause fields or smaller, highly variable with time on the scale of the Juno orbit, and of unknowable orientation. We thus conclude that it is justified to neglect this contribution on this basis, with the inclusion of the following text.

"For related reasons we also neglect the penetrating IMF term in equation (2), which is unknown when Juno is inside the magnetosphere, highly variable in direction with time, and typically of magnitude ~0.1-1 nT (Nichols et al., 2006, 2017). This field too, with penetration coefficient $k < 1$, is therefore similarly negligible in the $r < 60$ RJ middle magnetosphere studied here. "
* * *
*paragraph starting l. 12, p. 4: the discussion on the sensibility of S to the range of measurements considered for the fit is of prime importance. It needs some clarification and also expanded to justify the choice of range.*
* * *
The choice of ranges for analysis of the Juno data now considered in the revised paper is fully explained on page 6 lines 12–14 and page 7 lines 1–7 (plus Table 1) as follows.

"With regard to the choice of interval employed to minimize $S$, we note that use of data from the innermost region is not optimal. The JRM09 internal planetary field model differs from observations at periapsis (1.06 RJ ) by $0.3 \times 10^5$ nT (Connerney et al., 2018), which is reasonable accuracy for describing an observed field of magnitude ~$8 \times 10^5$ nT, but does not allow us to distinguish the magnetodisc field of order 100 nT on this background. We thus restricted the inner border of the interval to consider $r > 5$ R$_J$ only. However, on most passes examined here, the inner radial limit is set instead at somewhat larger radii by the data that are presently available for study. A further limitation on the region of calculation of $S$ in the outer magnetosphere arises from the fact that the paraboloid model does not display regions of low field strength during intersections with the magnetodisc, as is observed in the field at larger distances, due to the use of the infinitely thin disc approximation (see Section 4). It is thus necessary to avoid these regions by also setting a maximum radial distance, $R_{max}$, on each pass (see Figure 2 for perijove 1)."
* * *
*l. 7, p. 5: what does the value S = 0.2 correspond to concretely in terms of statistics? As it stands it seems to be an arbitrary choice.*

*l. 14, p. 5: I am not sure to follow the argument. What is it meant by 'acceptable pairs of parameters are aligned with the line to some extent'? Some clarification needed.*
* * *
Since the method of parameter optimization has now been changed as indicated above, and the corresponding text and figure omitted, these comments are no longer relevant to the revised paper.
* * *
*l. 1-5, p. 6: the discussion about the discrepancies observed in the internal field is too vague and lack content.*
* * *
At page 6 lines 13-14 in the revised version we simply report factually on the accuracy with which the published JRM09 internal field model agrees with the published periapsis data on Juno PJ-01.
* * *
*paragraph starting l. 2, p. 7: as mentioned before does it really make sense to take arbitrary values for the CAN disc parameters. Wouldn't it make more sense to carry a proper fit?*
* * *
As indicated above, a full fit and comparison with the CAN model is now presented in Figure 7.
* * *
*l. 7, p. 8: what do you mean by 'magnetodisc models with azimuthal current dependencies different from $r-2$ should also be investigated'? The CAN disc model just used in that Section varies as $r-1$. Do you have any suggestion? In Achilleos, Guio and Arridge (2010), it is suggested that the dependency is steeper than $r-1$.*
* * *
According to our results the dependence is steeper than $r^{-1}$, but less steep than $r^{-2}$. Further analysis is the topic of on-going research.
* * *
**Ivan A Pensionerov on behalf of the co-authors**

**27 November 2018**

[revised manuscript text omitted]
}} \mid R_{\mathrm{max}}$, $R_{\mathrm{J}}$ inbound | $R_{\mathrm{min}} \mid R_{\mathrm{max}}$, $R_{\mathrm{J}}$ outbound |
|---|---|---|---|---|---|
| Ulysses | 2.50 | 18.4 | 92 | | |
| PJ-00 | 2.57 | 18.6 | 95 | not available | 31.5 \| 60 |
| PJ-01 | 2.77 | 12.3 | 95 | 5.0 \| 45 | 5.0 \| 60 |
| PJ-02 | 2.67 | 13.7 | 95 | 13.3 \| 40 | not available |
| PJ-03 | 2.75 | 14.3 | 95 | 16.5 \| 40 | 8.9 \| 60 |
| PJ-04 | 2.43 | 14.0 | 95 | 13.7 \| 35 | 12.3 \| 60 |
| PJ-05 | 2.33 | 13.4 | 95 | 10.6 \| 30 | 10.5 \| 60 |
| PJ-06 | 2.31 | 12.5 | 95 | 8.0 \| 20 | 17.2 \| 60 |
| PJ-07 | 2.49 | 12.4 | 95 | not usable | 19.7 \| 60 |
| PJ-08 | 2.38 | 13.1 | 95 | not usable | 19.5 \| 60 |
| PJ-09 | 2.26 | 10.7 | 95 | not usable | 8.3 \| 60 |

With regard to the choice of interval employed to minimize $S$, we note that use of data from the innermost region is not optimal. The JRM09 internal planetary field model differs from observations at periapsis ($1.06\,R_{\mathrm{J}}$) by $0.3\cdot10^5\,\mathrm{nT}$ (Connerney et al., 2018), which is  reasonable accuracy for describing  an observed field of  magnitude $\sim 8\cdot10^5\,\mathrm{nT}$, but does not allow us to distinguish the magnetodisc field  of order $100$ nT on this background.

5 We thus restricted the inner border of the interval to consider ~~only $r > 5\,R_{\mathrm{J}}$. This is an arbitrary value, but the specific position within a range $\sim 5 - 10\,R_{\mathrm{J}}$ of the inner border of the fitting interval does not significantly affect the location of the minimum in $S$. On the other hand, the location of the minimum of the root-mean-square absolute deviation does depend strongly on the position of the inner fitting interval boundary, which is another reason not to use it for the present problem~~ $r > 5\,R_{\mathrm{J}}$ only. However, on most passes examined here, the inner radial limit is set instead at somewhat larger radii by the data that are

10 presently available for study. A further limitation on the region of calculation of $S$ in the outer magnetosphere arises from the fact that the paraboloid model does not display regions of low field strength during intersections with the magnetodisc, as is observed in the field at larger distances, due to the use of the infinitely thin disc approximation (see Section 4).  It is thus necessary to avoid these regions by also setting a maximum radial distance, $R_{\mathrm{max}}$, on each pass (see Figure 2 for perijove 1).

15  We thus minimize $S$ in the  inbound and outbound radial ranges between $R_{\mathrm{min}}$ and  $R_{\mathrm{max}}$ on each pass to determine the best fit magnetodisc

[Figure]

**Figure 5.** Observed (black) and modelled (red) residual fields in JSM cylindrical components, together with the residual field magnitude, for Juno perijove 1. The residual field is the observed field with the JRM09 internal field subtracted. The fields are plotted versus spherical radial distance with inbound data shown on the left and outbound data on the right. The same model field is used for both.

parameters. The minimization was undertaken using the Trust Region Reflective procedure (Branch et al., 1999). The best fit values are given, together with the radial ranges employed, in Table 1, where we also compare with the values derived by Alexeev and Belenkaya (2005) from Ulysses inbound data. For all the Juno fits we found that the best fit outer disc radius $R_{\mathrm{DC1}}$ was the maximum value of 95 $R_J$ allowed in the fitting process, set by requiring that the disc radius should be less than the subsolar magnetopause radius (100 $R_J$) by a few $R_J$. This indicates that the current density in the model disc, varying as $r^{-2}$, decreases somewhat too quickly with distance. The values of the inner disc radius $R_{\mathrm{DC2}}$  lie between 10.7 and 18.6 $R_J$, usually smaller than the value of 18.4 $R_J$, derived from the Ulysses data, while the field strength parameter $B_{\mathrm{DC}}$  Resulting intervals for the two fitted parameters are then found

[Figure]

**Figure 6.** As for Figure 5, but for perijove 6.

to be as follows,  varies between 2.3 and  ,2.8 nT, similar to the

$$M_{MD} = \frac{B_{DC}}{2} R_{DC1}^3 \left(1 - \frac{R_{DC2}}{R_{DC1}}\right)$$

5  Ulysses value of 2.5 nT.

[revised manuscript text omitted]